# Power analysis of transcriptome-wide association study: Implications for practical protocol choice

Chen Cao[1‡], Bowei Ding[2‡], Qing Li[1], Devin Kwok[2], Jingjing Wu[2*], Quan Long[1,2,3,4*]

**1** Department of Biochemistry & Molecular Biology, Alberta Children's Hospital Research Institute, University of Calgary, Calgary, Canada, **2** Department of Mathematics & Statistics, University of Calgary, Calgary, Canada, **3** Department of Medical Genetics, University of Calgary, Calgary, Canada, **4** Hotchkiss Brain Institute, O'Brien Institute for Public Health, University of Calgary, Calgary, Canada

‡ These authors share first authorship on this work.
* jinwu@ucalgary.ca (JW); quan.long@ucalgary.ca (QL)

**Data Availability Statement:** All relevant data are within the manuscript and its Supporting Information files.

**Funding:** JW is supported by an NSERC Discovery Grant (RGPIN-2018-04328). QL is supported by an

## Abstract

The transcriptome-wide association study (TWAS) has emerged as one of several promising techniques for integrating multi-scale 'omics' data into traditional genome-wide association studies (GWAS). Unlike GWAS, which associates phenotypic variance directly with genetic variants, TWAS uses a reference dataset to train a predictive model for gene expressions, which allows it to associate phenotype with variants through the mediating effect of expressions. Although effective, this core innovation of TWAS is poorly understood, since the predictive accuracy of the genotype-expression model is generally low and further bounded by expression heritability. This raises the question: to what degree does the accuracy of the expression model affect the power of TWAS? Furthermore, would replacing predictions with actual, experimentally determined expressions improve power? To answer these questions, we compared the power of GWAS, TWAS, and a hypothetical protocol utilizing real expression data. We derived non-centrality parameters (NCPs) for linear mixed models (LMMs) to enable closed-form calculations of statistical power that do not rely on specific protocol implementations. We examined two representative scenarios: causality (genotype contributes to phenotype through expression) and pleiotropy (genotype contributes directly to both phenotype and expression), and also tested the effects of various properties including expression heritability. Our analysis reveals two main outcomes: (1) Under pleiotropy, the use of predicted expressions in TWAS is superior to actual expressions. This explains why TWAS can function with weak expression models, and shows that TWAS remains relevant even when real expressions are available. (2) GWAS outperforms TWAS when expression heritability is below a threshold of 0.04 under causality, or 0.06 under pleiotropy. Analysis of existing publications suggests that TWAS has been misapplied in place of GWAS, in situations where expression heritability is low.

NSERC Discovery Grant (RGPIN-2017-04860), a Canada Foundation for Innovation JELF grant (36605), a New Frontiers in Research Fund (NFRFE-2018-00748), a Clinical Research Fund (10027289) and a Startup grant (10013532) supported by Alberta Children's Hospital Research Institute (ACHRI). CC is supported by ACHRI postdoctoral scholarship. The funders had no role in study design, data collection and analysis, decision to publish, or preparation of the manuscript.

**Competing interests:** The authors have declared that no competing interests exist.

## Author summary

We compared the effectiveness of three methods for finding genetic effects on disease in order to quantify their strengths and help researchers choose the best protocol for their data. The genome-wide association study (GWAS) is the standard method for identifying how the genetic differences between individuals relate to disease. Recently, the transcriptome-wide association study (TWAS) has improved GWAS by also estimating the effect of each genetic variant on the activity level (or expression) of genes related to disease. The effectiveness of TWAS is surprising because its estimates of gene expressions are very inaccurate, so we ask if a method using real expression data instead of estimates would perform better. Unlike past studies, which only use simulation to compare these methods, we incorporate novel statistical calculations to make our comparisons more accurate and universally applicable. We discover that depending on the type of relationship between genetics, gene expression, and disease, the estimates used by TWAS could be actually more relevant than real gene expressions. We also find that TWAS is not always better than GWAS when the relationship between genetics and expression is weak and identify specific turning points where past studies have incorrectly used TWAS instead of GWAS.

This is a *PLOS Computational Biology* Methods paper.

## Introduction

High-throughput sequencing instruments have enabled the rapid profiling of transcriptomes (RNA expression of genes) [1–4], proteomes (proteins) [5–7] and other 'omics' data [8–10]. These 'omics' provide insight into the intermediary effects of genotypes on endophenotypes, and can improve the ability of genome-wide association studies (GWAS) to find associations between genetic variants and disease phenotypes. [11–13]. The integration of diverse 'omics' data sources remains a challenging and active field of research [14–17].

One approach to integrating 'omics' and GWAS is the transcriptome-wide association study (TWAS), which quantitatively aggregates multiple genetic variants into a single test using transcriptome data. Pioneered by Gamazon *et al* [18], the TWAS protocol typically has two steps. First, a model is trained to predict gene expressions from local genetic variants near the focal genes, using a reference dataset containing both genotype and expression data. Second, the pretrained model is used to predict expressions from genotypes in the association mapping dataset under study, which contains genotypes and phenotypes (but not expression). The predicted expressions are then associated to the phenotype of interest. TWAS can also be conducted with summary statistics from GWAS datasets (i.e. meta-analysis) as first demonstrated by Gusev *et al.* [19,20]. TWAS has since achieved significant popularity and success in identifying the genetic basis of complex traits [21–27], inspiring similar protocols for other endophenotypes such as IWAS for images [28] and PWAS for proteins [29].

Despite its demonstrated effectiveness, important questions remain regarding the theoretical conditions under which TWAS is superior to GWAS. **First**: TWAS mapping relies entirely on predicted expressions, but as shown by many methodological papers, the mean $R^2$ between predicted and actual expressions is very low (around 0.02 ~ 0.05). This is in part due to low expression heritability [18], which bounds the maximum predictive accuracy attainable by the genotype-expression model. Naturally, one can ask: given sufficiently low expression

heritability, is there is a point at which TWAS performs worse than GWAS? Indeed in real data, genes discovered with significant TWAS $p$-values tend to have a higher $R^2$, and thus expression heritability, than on average [18,19,30–32]. We therefore investigate the effect of expression heritability on the power of TWAS, as well as its interactions with trait heritability, phenotypic variance from expressions, number of causal genes, and genetic architecture. **Second**: as described by Gamazon et al. [18], the key insight of TWAS is that it aggregates sensible genetic variants to estimate "genetically regulated gene expression", or GReX [18], for use in downstream GWAS. Given this hypothesis, one may ask if actual expression data would further improve the power of downstream GWAS over predicted expressions. This is not a trivial question, as although actual expressions do not suffer from prediction errors, they also include experimental or environmental noise which masks the genetic component of expression. To test this problem, we invent a hypothetical protocol associating real expressions to phenotype, which we call "expression mediated GWAS" or emGWAS. While emGWAS is not in practical use due to the difficulties of accessing relevant tissues (e.g., in the studies of brain diseases), it can potentially be applied to future analyses of diseases where tissues are routinely available (e.g., blood or cancerous tissues). More importantly, emGWAS serves as a useful benchmark for evaluating the theoretical properties of TWAS-predicted expressions against ground truth expression data. By analyzing the power of TWAS, GWAS, and emGWAS, we develop practical guidelines for choosing each protocol given different expression heritability and genetic architectures.

While there has been an existing study comparing the power of GWAS, TWAS, and a protocol which integrates eQTLs with GWAS [33], the existing study is purely simulation-based, whereas we determine power directly using traditional closed-form analysis. We derive non-centrality parameters (NCPs) for the relevant statistical tests and the linear mixed model (LMM) in particular (**Methods**). Our derivation uses a novel method to convert an LMM into a linear regression by decorrelating the covariance structure of the LMM response variable (**Methods**). To our best knowledge, this is the first closed-form derivation of the NCP for LMMs in current literature, with potential for broad applications as LMMs are the dominant models used in GWAS and portions of the TWAS pipeline.

Unlike pure simulations, which stochastically resample the alternative hypothesis to estimate statistical power, our closed-form derivation directly calculates power from a particular configuration of association mapping data. As a result, our method saves computational resources, yields more accurate power estimations, and adapts easily to similar protocols such as IWAS [28] and PWAS [29,34]. Moreover, as the closed-form derivation avoids conducting the actual regression, our power calculations do not depend on specific implementations of GWAS and TWAS, which could otherwise cause our results to vary due to differences in filtering inputs or parameter optimizations. Our work therefore characterizes the theoretical power of the protocols across all LMM-based implementations and datasets, although we are unable to account for power losses due to practical implementation issues.

In the following section we describe our novel derivation of NCPs for LMMs and our power analyses of GWAS, TWAS, and emGWAS. We present guidelines on the applicability of each protocol under different input conditions and discuss potential limitations of our approach as well as areas for future research.

## Methods

### Mathematical definitions of GWAS, TWAS, and emGWAS protocols

While there are many variations of GWAS and TWAS [18,19,35–39], in this work we assume that multiple genes contribute to phenotypic variation, and for each causal gene, multiple

single nucleotide polymorphisms (SNPs) contribute to both gene expression and phenotype. This setting is motivated by the fact that most complex traits are known to have multiple contributing loci, and TWAS fundamentally assumes that genes have multiple local causal variants. To ensure consistency, we apply the same assumptions in the design of the hypothetical protocol emGWAS. Specifically, we define the following models:

**GWAS.** For GWAS, we adopted a standard LMM similar to EMMAX [35]

$$Y = \beta_{j0}\mathbf{1} + \beta_{j1}X_j + u + \varepsilon, \quad j = 1, 2, \ldots, n_x, \tag{1}$$

where $n$ is the number of individuals, $n_x$ is the total number of genetic variants, $Y$ is an $n{\times}1$ vector of phenotypes, $\mathbf{1}$ is an $n{\times}1$ vector of ones, $X_j$ is an $n{\times}1$ genotype vector with $X_{ij} \in \{0,1,2\}$ representing the number of minor allele copies for the $i^{th}$ individual and $j^{th}$ genetic variant, $\beta_{j0}$ and $\beta_{j1}$ are the intercept and effect size of the genetic variant, $u$ is an $n{\times}1$ vector of random effects following the multivariate normal distribution, i.e. $u \sim N(0, \sigma_g^2 K_x)$, and $\varepsilon$ is an $n{\times}1$ vector of errors with $\varepsilon \sim N(0, \sigma_e^2 I)$. In the distributions of $u$ and $\varepsilon$, $\sigma_g^2$ and $\sigma_e^2$ are their respective variance components, $I$ is an $n{\times}n$ identity matrix, and $K_x$ is the genomic relationship matrix (GRM), which is a known $n{\times}n$ real symmetric matrix. Following Patterson *et al* [40], $K_x$ is calculated by

$$K_x = \frac{1}{n_x}\tilde{X}\tilde{X}^T, \tag{2}$$

where $n_x$ is the total number of genetic variants and $\tilde{X}$ is a standardized $n{\times}n_x$ matrix. For example, an element $\tilde{X}_{ij}$ in the $j^{th}$ genetic variant column is calculated as

$$\tilde{X}_{ij} = \frac{X_{ij} - \bar{X}_{.j}}{S_{Xj}}, \tag{3}$$

where $\bar{X}_{.j} = \frac{1}{n}\sum_{i=1}^{n} X_{ij}$ and $S_{Xj}^2 = \frac{1}{n-1}\sum_{i=1}^{n}(X_{ij} - \bar{X}_{.j})^2$ are the sample mean and sample variance of the $j^{th}$ variant, respectively.

**emGWAS.** For emGWAS, we first regress the phenotype on the actual (not predicted) expressions, and then regress the expressions on individual local genetic variants in a similar manner as a cis-eQTL analysis. We chose the LMM to associate phenotype with expression, since under the assumption that multiple genes contribute to phenotype, we expect that the random term of the LMM can capture the effects of non-focal genes. We calculate the GRM from DNA instead of expressions because they provide better estimates of pairwise relationships between study participants than correlations based on predicted expression data. We chose to use linear regression (LM) to model the association between expression and local genetic variants (which correspond to cis-eQTLs), as it is the most common model used in cis-eQTL analyses.

Specifically, the phenotype-expression model is

$$Y = \beta_{l0}\mathbf{1} + \beta_{l1}Z_l + u + \varepsilon, \; l = 1, 2, \ldots, n_z, \tag{4}$$

where $n$, $Y$, $\mathbf{1}$, $u$ and $\varepsilon$ have identical interpretations as in the GWAS model from (1), $n_z$ is the total number of genes, $Z_l$ is an $n{\times}1$ gene expression vector for the $l^{th}$ gene, and $\beta_{l0}$ and $\beta_{l1}$ are the intercept and effect size of the gene.

The linear regression associating gene expression with local genetic variants is

$$Z_l = \beta_{lk0}\mathbf{1} + \beta_{lk1}X_{lk} + \varepsilon_{el}, \; l = 1, 2, \ldots, n_z, k = 1, 2, \ldots, n_{el}, \tag{5}$$

where $X_{lk}$ is an $n_{el}{\times}1$ vector of the $k^{th}$ local genetic variants for the $l^{th}$ gene, $\varepsilon_{el} \sim N(0, \sigma_{el}^2 I)$ is a

$n \times 1$ vector of errors with variance component $\sigma^2_{el}$, $n_{el}$ is the total number of local genetic variants in the $l^{th}$ gene, and $\beta_{lk0}$ and $\beta_{lk1}$ are the intercept and effect size of the variant.

**TWAS.**   For TWAS, we apply an analysis similar to emGWAS, except that gene expressions are predicted using a pretrained elastic-net model. Specifically,

$$Y = \beta_{Pl0}\mathbf{1} + \beta_{Pl1}\widehat{Z}_l + u + \varepsilon, \quad l = 1, 2, \ldots, n_z, \tag{6}$$

where $\widehat{Z}_l$ is the altered notation representing an $n \times 1$ vector of predicted gene expressions for the $l^{th}$ gene, and $\beta_{Pl0}$ and $\beta_{Pl1}$ are the intercept and effect size of the predicted gene expression.

There are several methods to estimate gene expression including least absolute shrinkage and selection operator (LASSO) and elastic-net. Gamazon *et al.* has shown that elastic-net has good performance and is more robust to minor changes in the input variants [18]. We therefore use the "glmnet" package in R to train a predictive model using elastic-net. The objective function in "glmnet" is

$$L_{enet}(\beta) = \frac{1}{2n}\|Z - X\beta\|^2 + \lambda\left(\frac{1-\alpha}{2}\|\beta\|^2 + \alpha\|\beta\|_1\right) \tag{7}$$

where $\lambda$ and $\alpha$ are tuning parameters. The penalty term is a convex (linear) combination of LASSO and ridge penalties, where $\alpha = 1$ is equivalent to the LASSO objective function, and $\alpha = 0$ is equivalent to ridge regression. Optimal values of $\lambda$ and $\alpha$ were chosen by minimizing the cross-validated squared-error. Readers are referred to **Appendix A in S1 Text** for details.

In practice, the specific regression model varies depending on the tool in use. For example, the leading TWAS tool PrediXcan [18] does not include the random effects of a mixed model, and many TWAS tools can also analyze summary statistics instead of subject-level genotypes [19]. The motivation of this work is to reveal the key issues of using gene expressions as mediations, therefore has to adapt comparable framework. In other words, we do not intend to compare LMM against linear regression, which will mislead the comparison between GWAS and TWAS. Since LMMs are dominant in GWAS, we chose LMMs as the underlying model for all of the protocols we analyze, which allows us to compare them under an equivalent statistical framework. We believe that LMMs are a sensible approach for TWAS, since the random term can capture the genetic contributions of non-focal genes.

## Closed-form derivation of NCP and power calculation

The non-centrality parameter (NCP) measures the distance between a non-central distribution and a central distribution under a specific alternative hypothesis. The NCP enables calculation of the probability of rejecting the null hypothesis, assuming the central distribution, when the alternative hypothesis is correct. As such, the NCP naturally allows the power of a statistical test to be determined in a closed form. We have developed the following method to derive the NCP for LMMs, which we believe is new to the literature.

For a standard simple linear regression, the NCP of a $t$-test of the coefficient of the predictor variable can be derived similarly to a one-sample $t$-test statistic as follows: if $X_1, \ldots, X_n \sim N(\mu, \sigma)$ is a simple random sample, then the one-sample $t$-test statistic for evaluating the null hypothesis $H_0: \mu = \mu_0$ is

$$T = \frac{\bar{X} - \mu_0}{\frac{S}{\sqrt{n}}} = \frac{\frac{\sqrt{n}(\bar{X} - \mu_0)}{\sigma}}{\sqrt{\frac{(n-1)S^2}{\sigma^2}}{n-1}}} \sim t_{n-1}, \tag{8}$$

where $\bar{X}$ and $S$ are the sample mean and (unbiased) sample standard deviation respectively.

Under $H_0$, $\sqrt{n}(\bar{X} - \mu_0)/\sigma \sim N(0, 1)$ and $(n - 1)S^2/\sigma^2 \sim \chi^2_{n-1}$, and thus $T \sim t_{n-1}$. Under the alternative hypothesis $H_a$:$\mu = \mu_a$, the test statistic $T = \frac{\sqrt{n}[(\bar{X} - \mu_a) + (\mu_a - \mu_0)]/\sigma}{\sqrt{\frac{(n-1)S^2/\sigma^2}{n-1}}}$ follows a non-central $t$ distribution with NCP given by

$$\nu = \frac{\mu_a - \mu_0}{\sigma/\sqrt{n}} \tag{9}$$

To derive a closed-form NCP for LMMs, we convert the LMM to a linear regression without intercept by decorrelating the response variable and the predictors, a technique that has previously been applied to mixed models [41,42]. The procedure is as follows: we first fit the null model $Y_c = u + \varepsilon$ with no genetic variants, following an existing innovation for reducing the computational cost of repeatedly factorizing the GRM when analyzing many variants [35,42]. We then estimate $\sigma^2_g$ using the Newton-Raphson method detailed in **Appendix B in S1 Text**. Denoting the eigen decomposition of the GRM as $K_x = U_x \Lambda_x U_x^{-1}$, we construct the de-correlation matrix as

$$D_x = (\sigma^2_g \Lambda_x + \sigma^2_e I)^{-\frac{1}{2}} U_x^T. \tag{10}$$

By left multiplying both $X$ and $Y$ by $D_x$, and denoting $X^* = D_x X = (X_1^*, X_2^*, \ldots, X_n^*)^T$ and $Y^* = D_x Y = (Y_1^*, Y_2^*, \ldots, Y_n^*)^T$, the covariance structure in $Y^*$ is thus removed and a linear regression of $Y^*$ on $X^*$ is equivalent to the original LMM model. A proof of the validity of this decorrelation structure is presented in **Appendix C in S1 Text**.

Based on the closed-form NCP for linear regression, we derive the estimated NCP of the LMM from (1), which is given by

$$\widehat{v}_{Gj} = \frac{\sum_{i=1}^n \widehat{X}_{ij}^* \widehat{Y}_i^* \sum_{i=1}^n \widehat{D}_{xi\cdot}^2 - \sum_{i=1}^n \widehat{Y}_i^* \widehat{D}_{xi\cdot} \sum_{i=1}^n \widehat{X}_{ij}^* \widehat{D}_{xi\cdot}}{\sqrt{\sum_{i=1}^n (\widehat{X}_{ij}^*)^2 (\sum_{i=1}^n \widehat{D}_{xi\cdot})^2 - (\sum_{i=1}^n \widehat{D}_{xi\cdot} \widehat{X}_{ij}^*)^2 \sum_{i=1}^n \widehat{D}_{xi\cdot}^2}}, \tag{11}$$

where $\widehat{X}_j^* = \widehat{D}_x X_j = (\widehat{X}_{1j}^*, \widehat{X}_{2j}^*, \ldots, \widehat{X}_{nj}^*)^T$, $\widehat{Y}^* = \widehat{D}_x Y = (\widehat{Y}_1^*, \widehat{Y}_2^*, \ldots, \widehat{Y}_n^*)^T$, and $\widehat{D}_{xi\cdot} = \sum_{j=1}^n \widehat{D}_{xij}$. A proof of this expression of the NCP for LMMs is in **Appendix D in S1 Text**.

The above result allows us to derive the statistical power of the GWAS, emGWAS, and TWAS protocols. For GWAS, we use the Bonferroni-corrected significance level $\alpha_x = \frac{0.05}{n_x}$ to account for multiple testing [43], where $n_x$ is the total number of SNPs. Throughout this paper, we use $f(t;v)$ to denote the probability density function of the non-central $t$ distribution with $n$-2 degrees of freedom and NCP $v$. The statistical power of the $j^{th}$ SNP can then be estimated by $P_{Gj} = \int_{F_0^{-1}(1-\alpha_x)}^{+\infty} f(t; \widehat{v}_{Gj}) dt$ using the estimated NCP $\widehat{v}_{Gj}$, where $F_0(t)$ is the cumulative distribution function of the central $t$ distribution with $n$-2 degrees of freedom, and $F_0^{-1}(1 - \alpha_x)$ gives the critical value for the central distribution. We directly implement this power computation in R via the function "pt", which takes the critical value, NCP, and degrees of freedom as parameters.

For emGWAS, we assume that the powers of the expression-phenotype and genotype-expression regression models (4) and (5) are independent of each other. For the model $Y = \beta_{l0}\mathbf{1} + Z_l\beta_{l1} + u + \varepsilon$ from (4), we left multiply the estimated $\widehat{D}_x$ to both sides of the equation so that the estimated NCP for the $l^{th}$ gene expression is given by

$$\widehat{v}_{eZl} = \frac{\sum_{i=1}^n \widehat{Z}_{il}^* \widehat{Y}_i^* \sum_{i=1}^n \widehat{D}_{xi\cdot}^2 - \sum_{i=1}^n \widehat{Y}_i^* \widehat{D}_{xi\cdot} \sum_{i=1}^n \widehat{Z}_{il}^* \widehat{D}_{xi\cdot}}{\sqrt{\sum_{i=1}^n (\widehat{Z}_{il}^*)^2 (\sum_{i=1}^n \widehat{D}_{xi\cdot}^2)^2 - (\sum_{i=1}^n \widehat{D}_{xi\cdot} \widehat{Z}_{il}^*)^2 \sum_{i=1}^n \widehat{D}_{xi\cdot}^2}}, \tag{12}$$

where $\widehat{Z}_l^* = \widehat{D}_x Z_l = (\widehat{Z}_{1l}^*, \widehat{Z}_{2l}^*, \ldots, \widehat{Z}_{nl}^*)^T$. We use the significance level $\alpha_z = \frac{0.05}{n_z}$ for each individual test, where $n_z$ is the total number of genes. The statistical power of detecting the $l^{th}$ gene expression is then estimated by $P_{eZl} = \int_{F_0^{-1}(1-\alpha_z)}^{+\infty} f(t; \widehat{v}_{eZl}) dt$. For the model from (5), we simply calculate the estimated NCP of the standard linear regression, which is

$$\widehat{v}_{eXlk} = \frac{\sum_{i=1}^n (X_{ilk} - X_{\cdot lk}) Z_{il}}{\sqrt{\sum_{i=1}^n (X_{ilk} - X_{\cdot lk})^2 \widehat{\sigma}_{el}}}, \tag{13}$$

where

$$\widehat{\sigma}_{el} = \frac{1}{n-2} \sum_{i=1}^n (Z_{il} - \bar{Z}_{\cdot l} + \widehat{\beta}_{lk}(X_{ilk} - \bar{X}_{\cdot lk}))^2. \tag{14}$$

Again, we use the significance level $\alpha_{el} = \frac{0.05}{n_{el}}$, where $n_{el}$ is the total number of local genetic variants in the $l^{th}$ gene, so that the power of detecting $X_{lk}$ is estimated by $P_{eXlk} = \int_{F_0^{-1}(1-\alpha_{el})}^{+\infty} f(t; \widehat{v}_{eXlk}) dt$. Since we assume the power of (4) and (5) are independent, the power of detecting the $l^{th}$ gene and the $k^{th}$ variants in the $l^{th}$ gene simultaneously is give by $P_{eZl} \cdot P_{eXlk}$. If the independence assumption is violated, i.e., the powers of these two steps are positively correlated, then the estimated power for emGWAS will be conservative.

For TWAS, the NCP is estimated in a similar manner as the first step of emGWAS, i.e.

$$\widehat{v}_{Tl} = \frac{\sum_{i=1}^n \widehat{\widehat{Z}}_{il}^* \widehat{Y}_i^* \sum_{i=1}^n \widehat{D}_{xi\cdot}^2 - \sum_{i=1}^n \widehat{Y}_i^* \widehat{D}_{xi\cdot} \sum_{i=1}^n \widehat{\widehat{Z}}_{il}^* \widehat{D}_{xi\cdot}}{\sqrt{\sum_{i=1}^n (\widehat{\widehat{Z}}_{il}^*)^2 (\sum_{i=1}^n \widehat{D}_{xi\cdot}^2)^2 - (\sum_{i=1}^n \widehat{D}_{xi\cdot} \widehat{\widehat{Z}}_{il}^*)^2 \sum_{i=1}^n \widehat{D}_{xi\cdot}^2}}, \tag{15}$$

where the only difference between (12) and (15) is that $\widehat{Z}_{il}^* = \widehat{D}_x Z_{il}$ in (15) is replaced by $\widehat{\widehat{Z}}_{il}^* = \widehat{D}_x \widehat{Z}_{il}$ in (15). The significance level is again $\alpha_z = \frac{0.05}{n_z}$ and the power is estimated by $P_{Tl} = \int_{F_0^{-1}(1-\alpha_z)}^{+\infty} f(t; \widehat{v}_{Tl}) dt$.

## Simulation of phenotype and expression

As the statistical power of each protocol depends on the magnitude of the genetic effect, we simulated input data at various effect sizes. While effect size depends on a combination of many factors, we chose to focus on the following three aspects. 1) We considered two genetic architectures: causality and pleiotropy (**Fig 1**). In the causality scenario, the contribution of genotype to phenotype is mediated through expression (**Fig 1A**), whereas in the pleiotropy scenario, genotype contributes to both expression and phenotype directly (**Fig 1B**). We did not consider the scenario where phenotype is causal to expression. 2) We considered the strength of three different variant components: trait heritability (the variance component of phenotype explained by genotype, denoted $h_{x=>y}^2$), expression heritability (the variance component of expression explained by genotype, denoted $h_{x=>z}^2$), and the phenotypic variance component explained by expression, denoted $h_{z=>y}^2$ and abbreviated as *PVX*. 3) We also considered the number of genes contributing to phenotype and the number of local genetic variants contributing to expression.

In all our simulations, we use real genotypes from the 1000 Genomes Project ($N = 2504$). Although there are multiple existing datasets containing both expressions and genotype, we chose to use simulated expressions instead as it is difficult to match real data exactly to desired properties such as expression heritability or the number of contributing genetic variants. By simulating expressions, we can perform a consistent power analysis across a comprehensive range of prespecified input conditions.

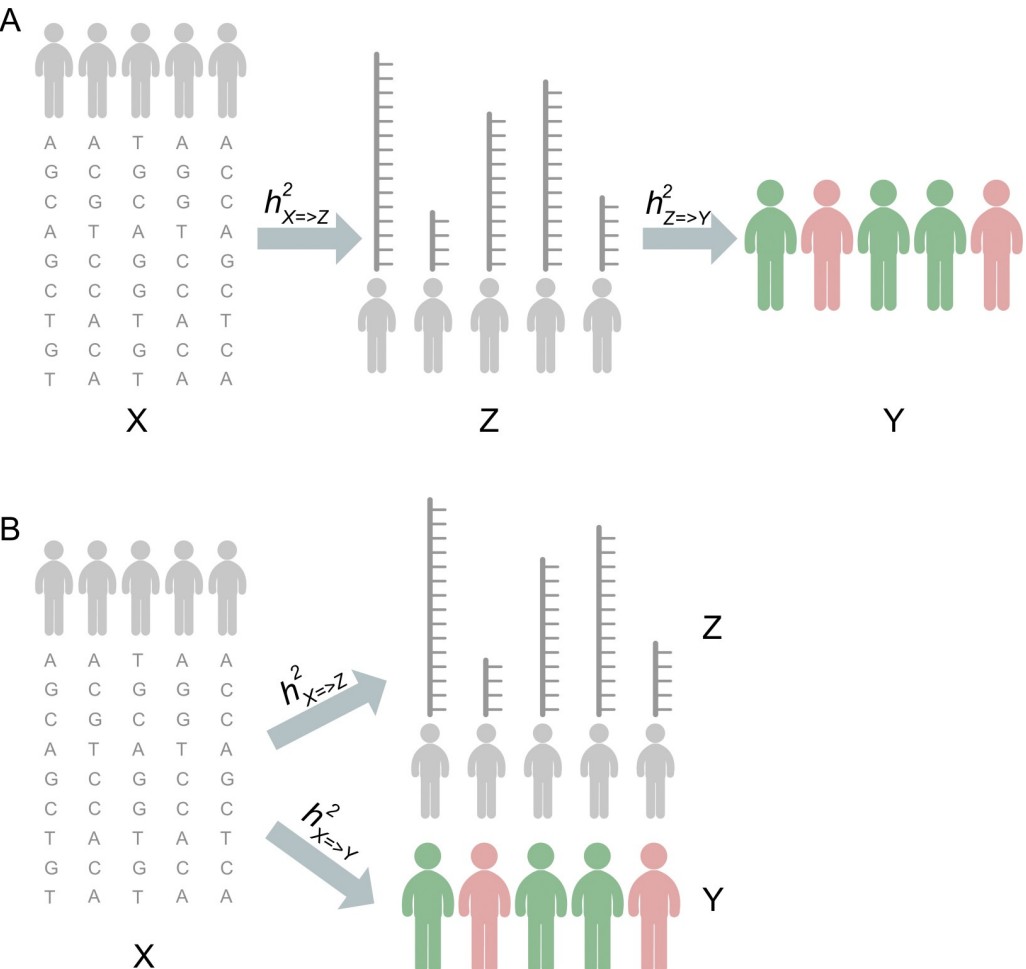

**Fig 1.** Causality (A) and Pleiotropy (B) scenarios for genotype (X), expression (Z) and phenotype (Y).

In the causality scenario, phenotypes were simulated with the following procedure. First, several genes ($n_{z-sig}$ = 4, 9, or 13) were selected as causal genes. For each gene (indexed by $l$ = 1,2,...,$n_{z-sig}$), several common and independent genetic variants were selected as causal variants ($n_{z(l)-sig}$ = 4 ~9, $MAF > 0.05$, and $R^2 < 0.01$). A linear combination of local variants in the $l^{th}$ gene is generated to produce the expression values $Z_{(l)}$, and a linear combination of these gene expressions **Z** is generated as the genomic contribution to phenotype. Note that at each step, we ensure the simulated linear combinations of variants and expressions match our desired values for expression heritability $h^2_{x=>z}$ and PVX $h^2_{z=>y}$ (**Appendix E in S1 Text**).

In the pleiotropy scenario, we followed a similar procedure except that the phenotype $Y$ was directly generated from a linear combination of genotypes, instead of expressions (**Appendix F in S1 Text**). Note that although the expressions $Z$ and phenotype $Y$ are unrelated by genuine biological causality, they are generated from the same genetic variants and are therefore statistically correlated. Therefore, if the trait heritability and expression heritability are sufficiently large, TWAS can still identify causal genes using the statistical correlation between genetic variants and expression.

We simulated both scenarios with expression heritability $h^2_{x=>z}$ from the values (2.5%, 3%, 4%, 6%, 8%, 10%, 30%), and with trait heritability $h^2_{x=>y}$ in the pleiotropy scenario or PVX

$h^2_{z=>y}$ in the causality scenario from the values (0.5%, 1%, 2.5%, 5%, 10%). Although we initially tested more extreme values, our **Results** show that the turning points where TWAS outperforms GWAS are well within the range of values presented here, and the relative performance of the protocols remains consistent under more extreme conditions. We therefore chose to restrict our discussion to the most relevant values for protocol selection, noting that the expression heritability values we examine are at the high-end of real observed values [18], while the trait heritability values are lower than typically found in GWAS.

Finally, as each simulation involves multiple variants and genes, the overall power of each protocol is defined as follows: the power of GWAS is the probability of detecting at least one causal variant in any causal gene, the power of emGWAS is the probability of detecting at least one gene and one local SNP of that gene simultaneously, and the power of TWAS is the probability that at least one predicted gene expression is significant. Specifically,

$$P_{GWAS} = 1 - \prod_{j=1}^{n_{x-sig}} (1 - P_{G(j)}), \tag{16}$$

$$P_{emGWAS} = 1 - \prod_{l=1}^{n_{z-sig}} (1 - P_{eZ(l)}P_{eX(l)}), \text{ where } P_{eX(l)} = 1 - \prod_{k=1}^{n_{z(l)-sig}} (1 - P_{eX(l)(k)}), \tag{17}$$

$$P_{TWAS} = 1 - \prod_{l=1}^{n_{z-sig}} (1 - P_{T(l)}), \tag{18}$$

where $n_{x-sig}$, $n_{z-sig}$ and $n_{z(l)-sig}$ denote the numbers of significant SNPs, genes, and SNPs in the $l^{th}$ significant gene respectively, $G(j)$ denotes the $j^{th}$ significant SNP identified by GWAS, $Z(l)$ and $X(l)(k)$ denote the $l^{th}$ significant gene and the $k^{th}$ significant SNP of the $l^{th}$ significant gene identified by emGWAS, and $T(l)$ denotes the $l^{th}$ significant gene identified by TWAS.

## Results

As a quality control measure, we compared the actual expression heritability and the mean $R^2$ of the predicted expressions (**Table 1**). As expected, the mean $R^2$ grows closer to the actual heritability value as expression heritability increases.

### Causality scenario

We first analyzed cases where expression heritability is high ($h^2_{x=>z} = 0.1$ or $0.3$) but the PVX is low (**Fig 2**). Overall, emGWAS clearly outperforms both GWAS and TWAS by a large margin,

**Table 1. Comparisons of $R^2$ of imputed gene expression under different levels of expression heritability and number of genetic variants.**

|  | Mean of $R^2$ | Sample Standard Deviation of $R^2$ |
|---|---|---|
| $h^2_1 = 0.025$ | 0.007847616 | 0.007415877 |
| $h^2_1 = 0.03$ | 0.01259302 | 0.008410582 |
| $h^2_1 = 0.04$ | 0.02319834 | 0.009481371 |
| $h^2_1 = 0.06$ | 0.04415579 | 0.01083593 |
| $h^2_1 = 0.08$ | 0.06465895 | 0.01175991 |
| $h^2_1 = 0.1$ | 0.08518152 | 0.01264175 |
| $h^2_1 = 0.3$ | 0.2886779 | 0.01514781 |

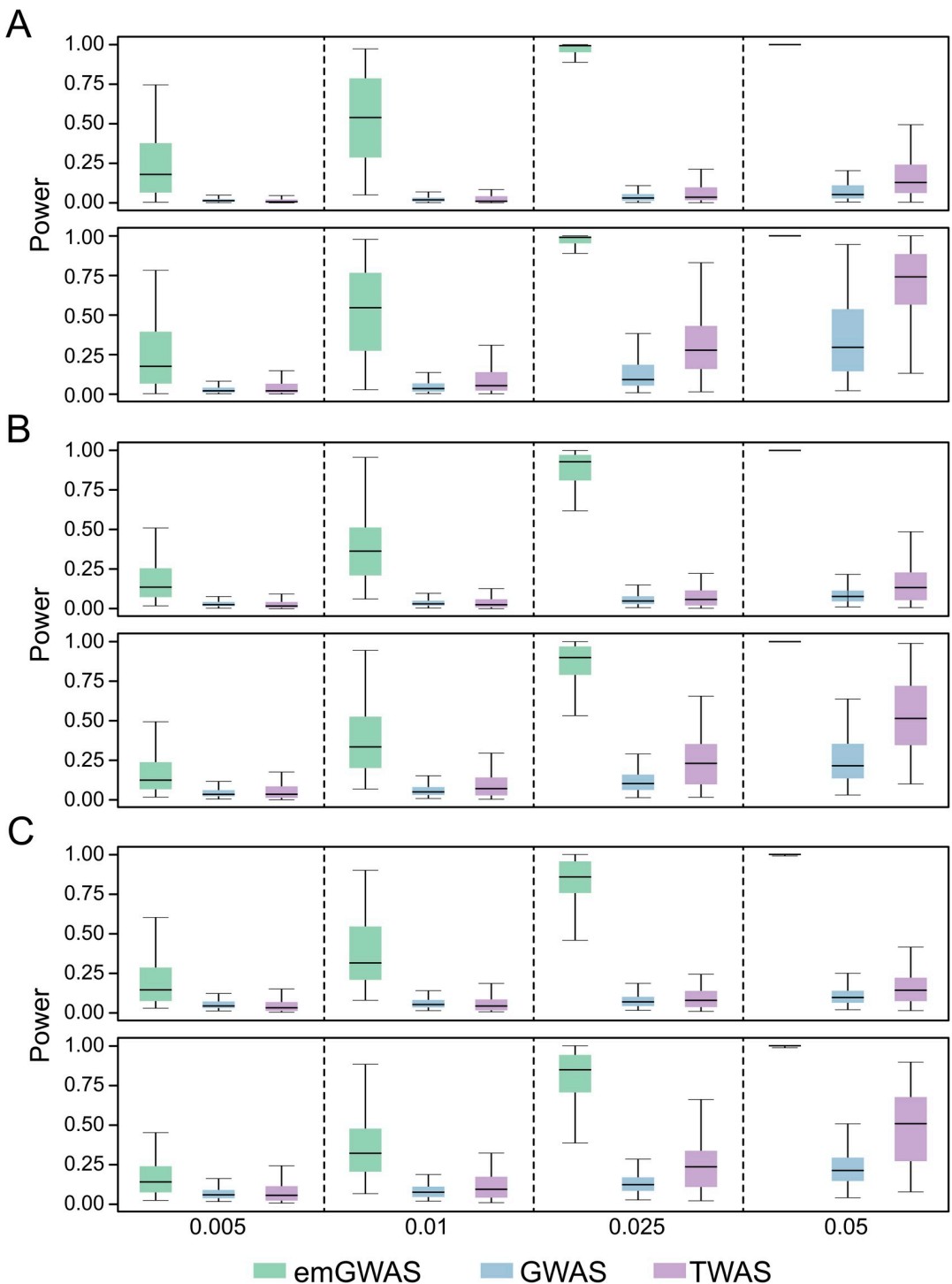

**Fig 2. Causality scenario when expression heritability is high and PVX is low.** The PVX is 0.005, 0.01, 0.025, and 0.05 in the four columns as indicated by the X-axis labels. The number of genes contributing to phenotype for (A), (B) and (C) are 4, 9, and 13 respectively. The expression heritability for the top and bottom rows of (A), (B) and (C) are 0.1 and 0.3 respectively. The number of causal variants per gene is randomly sampled from the interval [4,9].

and TWAS also generally outperforms GWAS. Note that although the PVX is low and favors GWAS, TWAS is still more powerful due to the high expression heritability, which shows that expression heritability affects the performance of TWAS more than the PVX. Consistent with intuition, we observed that GWAS and TWAS have higher power as expression heritability increases, whereas this increase is much smaller for emGWAS. The power of GWAS and emG-WAS reduces as the number of causal genes grows, whereas TWAS is largely unaffected by the number of causal genes. This is also consistent with intuition since TWAS uses GReX ($\widehat{Z}$) to aggregate genetic effects, avoiding the burden of multiple-testing correction.

We then analyzed cases where the PVX is high, but expression heritability is relatively low ($h^2_{x=>z} = 0.025, 0.03, 0.04$ or $0.08$). Evidently, emGWAS performs best with powers consistently at 1.0. The comparison between TWAS and GWAS is more nuanced, as TWAS is suboptimal to GWAS when the expression heritability is 0.025 or 0.03 (**Fig 3A and 3B**), begins to

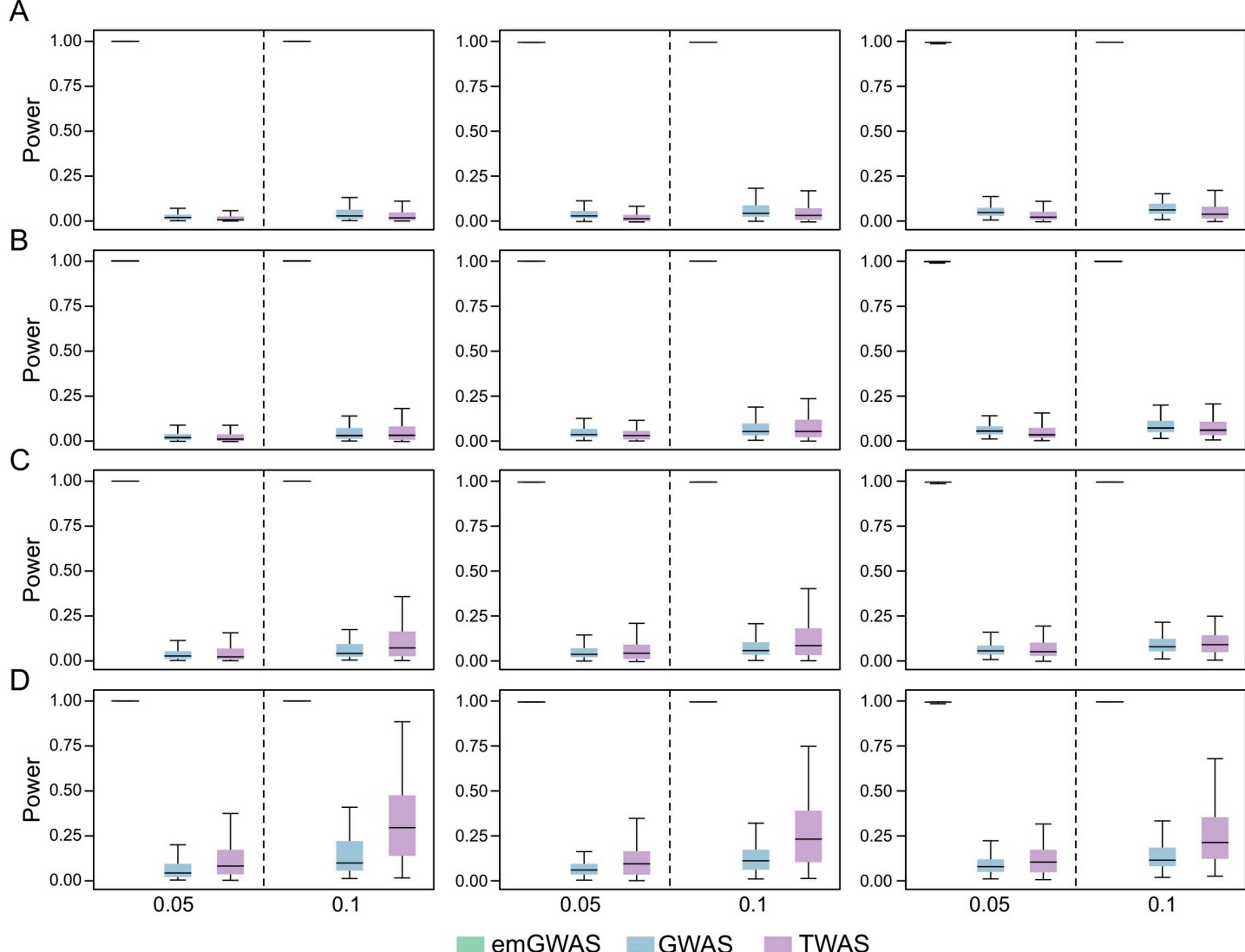

**Fig 3. Causality scenario when expression heritability is low and PVX is high.** The PVX is 0.05 and 0.1 in the two columns as indicated by the X-axis labels. The numbers of genes contributing to phenotype in the left, middle and right panels are 4, 9, and 13 respectively. The expression heritability levels in (A), (B), (C) and (D) are 0.025, 0.03, 0.04, and 0.08 respectively. The number of causal variants per gene is randomly sampled from the interval [4,9].

outperform GWAS when the expression heritability is 0.04 (**Fig 3C**), and clearly outperforms GWAS when the expression heritability is 0.08 (**Fig 3D**). This quantifies an important turning point in that GWAS is superior to TWAS when expression heritability is less than 0.04, even if PVX is high (favoring TWAS).

## Pleiotropy scenario

Again, we first analyze cases where expression heritability is high and trait heritability is low (**Fig 4**). Unlike in the causality scenario, the power of emGWAS is very low compared to TWAS and GWAS. A potential explanation is that when the effect of genetic variants on phenotype is not mediated through expressions, the non-genetic effects within the actual expressions add noise to emGWAS predictions. In contrast, the elastic-net model in TWAS captures only the genetic component of expressions, meaning the predicted expressions are a more accurate model of the direct genetic effect on phenotype. While errors are unavoidable in the elastic-net training process (as revealed in **Table 1**), our results show that the loss of power due to non-genetic effects is overwhelmingly greater than the loss due to training errors. As in the casualty scenario, TWAS generally outperforms GWAS except in the case where trait heritability is extremely low and the number of contributing genes is large, which is rare in practice. We therefore conclude that in both scenarios, TWAS has better power than GWAS when expression heritability is high.

We finally analyze cases where expression heritability is low but trait heritability is high. Here, emGWAS continues to be the least powerful of the three protocols. As in the causality scenario, we again observe a turning point where TWAS outperforms GWAS: TWAS has lower power than GWAS when the expression heritability is 0.025 or 0.04 (**Fig 5A and 5B**), TWAS has comparable power when the expression heritability is 0.06 (**Fig 5C**), and TWAS outperforms GWAS when the expression heritability is 0.08 (**Fig 5D**).

Our results can be summarized in two observations. First, emGWAS outperforms TWAS and GWAS in the casualty scenario, but is less powerful in the pleiotropy scenario regardless of the accuracy of the predicted expressions (**Table 1**). This demonstrates that when non-genetic components in expression do not contribute to phenotype (i.e. pleiotropy scenario), predicted expressions capture genetic contributions better than actual expressions (which include non-genetic components). Second, the turning point at which traditional GWAS outperforms TWAS is an expression heritability of less than 0.04 in the causality scenario, or 0.06 in the pleiotropy scenario.

These turning points are immediately relevant to the practical conduct of association mapping studies, as shown by the following analysis of expression heritability in existing TWAS publications. As few publications disclose their estimated expression heritability, we use published $R^2$ values of the correlation between predicted and actual expressions to approximate the underlying expression heritability. We use the difference between expression heritability and $R^2$ as calculated from our simulations (**Table 1**) to map these $R^2$ values to an estimated expression heritability (i.e. $R^2$ of 0.023 and 0.044 give expression heritability values 0.04 and 0.06, respectively), although in practice the true difference may vary depending on the predictive model used in each study. **Table 1** of the PrediXcan publication lists significant results from their paper, in which 14 out of 41 discovered genes have $R^2$ values less than 0.044, with 2 values less than 0.023. Additionally, our review of recent TWAS publications shows that most of the genes presented have mean $R^2$ values less than 0.044 or 0.023 (**Table 2**). As our power analysis indicated, GWAS may have better power than TWAS given these low expression heritability conditions. Although we are unable to determine if the genes discovered by these publications follow the causality or pleiotropy scenario, other advanced statistical models [44] may be used to determine appropriate thresholds to distinguish between pleiotropy and causality.

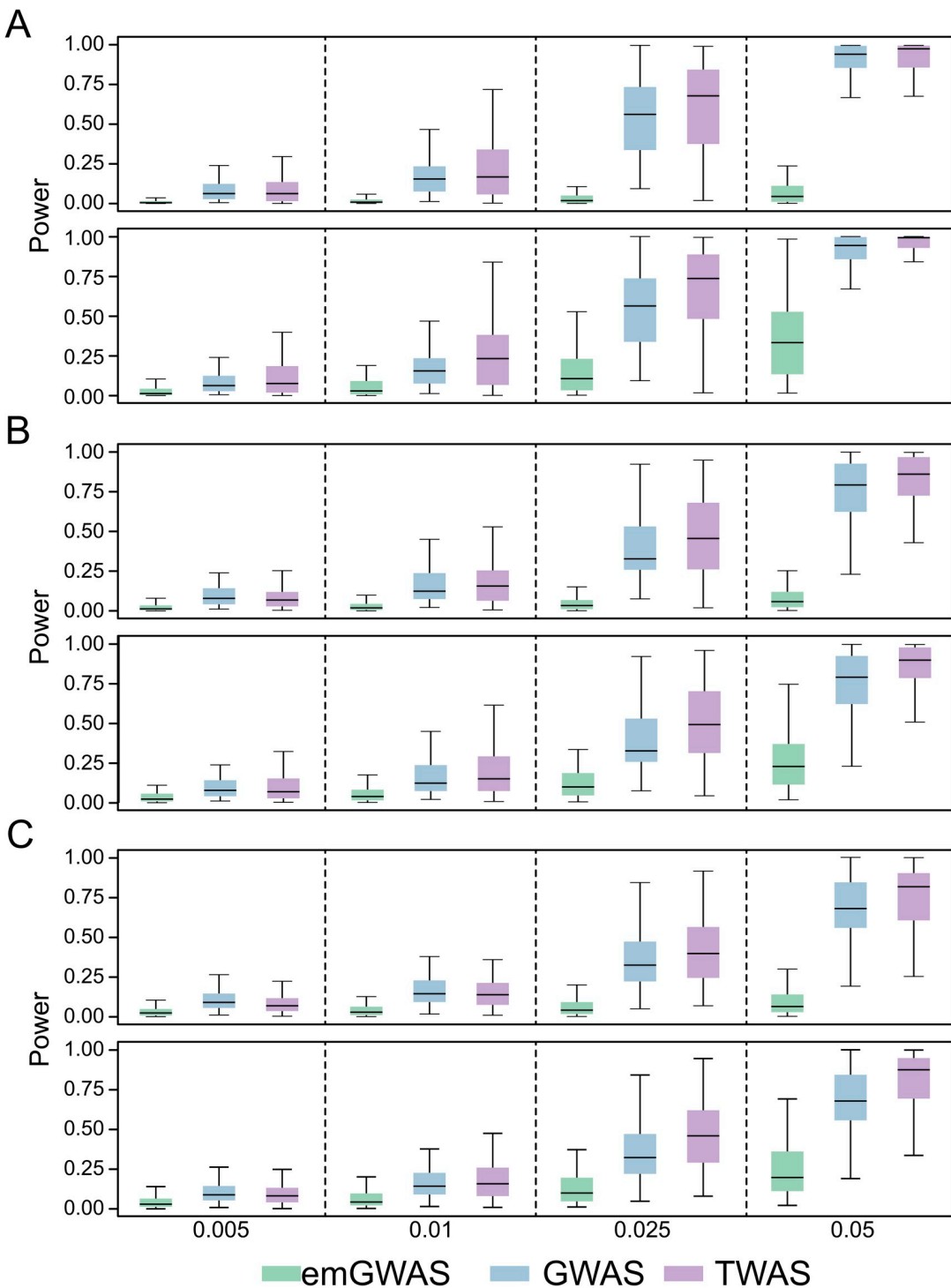

**Fig 4. Pleiotropy scenario when expression heritability is high and trait heritability is low.** The trait heritability is 0.005, 0.01, 0.025, and 0.05 in the four columns as indicated by the X-axis labels. The numbers of genes contributing to phenotype for (A), (B) and (C) are 4, 9, and 13 respectively. The expression heritability for the top and bottom rows of (A), (B) and (C) are 0.1 and 0.3 respectively. The number of causal variants per gene is randomly sampled from the interval [4,9].

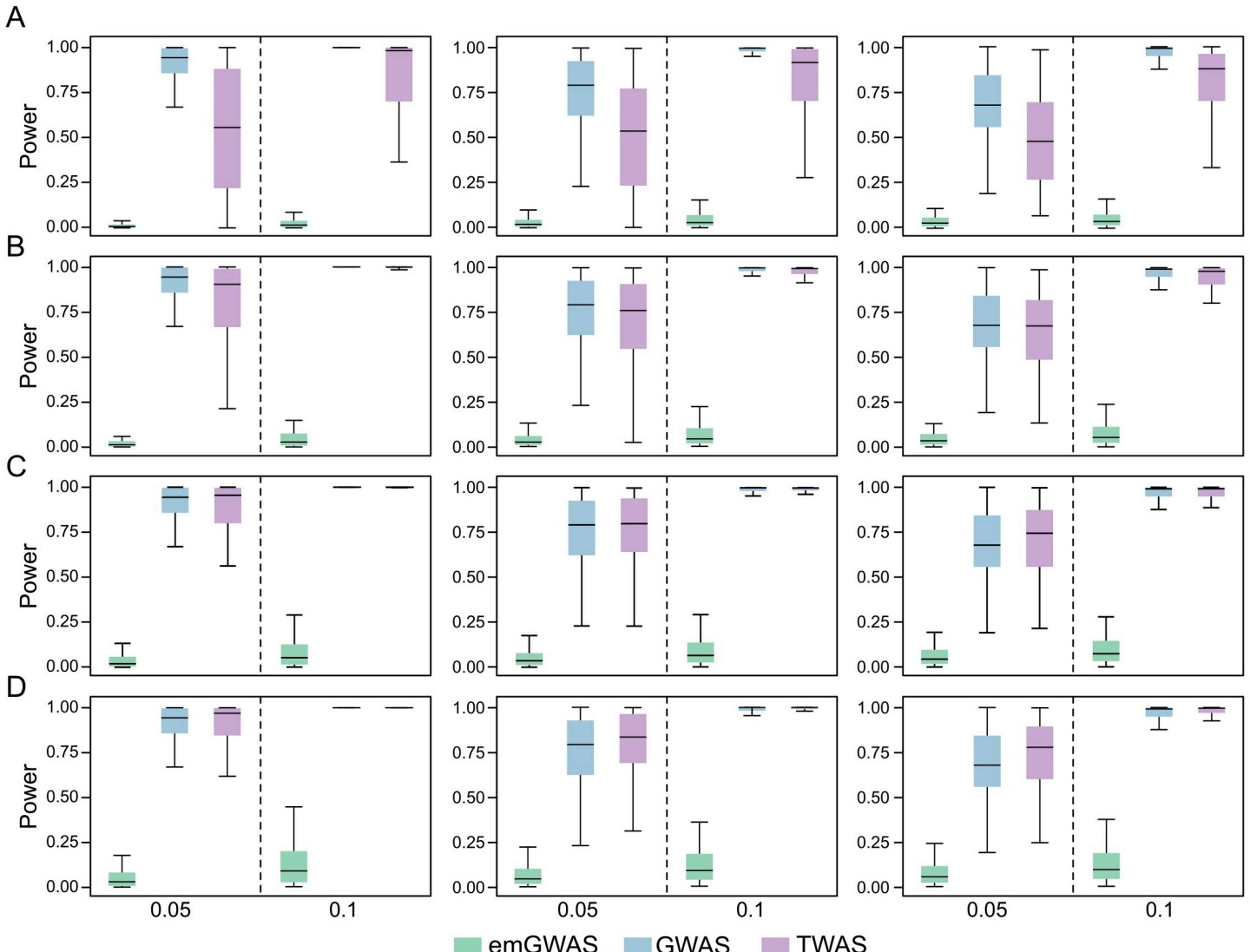

**Fig 5. Pleiotropy scenario when expression heritability is low and trait heritability is high.** The PVX is 0.05 and 0.1 in the two columns as indicated by the X-axis labels. The numbers of genes contributing to phenotype for the left, middle and right panels are 4, 9, and 13 respectively. The expression heritability levels in (A), (B), (C) and (D) are 0.025, 0.04, 0.06, and 0.08 respectively. The number of causal variants per gene is randomly sampled from the interval [4,9].

In summary, we suggest the following modifications to the TWAS protocol. First, one may estimate expression heritability in the reference panel and filter out genes with expression heritability less than 0.04. Second, after conducting TWAS association mapping, determine the underlying causality scenario (causality or pleiotropy) in order to choose an appropriate expression heritability threshold (0.04 or 0.06). Finally, conduct GWAS for each gene with an expression heritability below the given threshold.

## Application to the power estimation of EpiXcan

Our NCP-based framework can be applied to estimate the power of other protocols. To demonstrate this point, we estimated the power of EpiXcan [27], a novel TWAS-like protocol integrating epigenetic functional annotations to improve the accuracy of predicted expressions and therefore overall TWAS power. The original EpiXcan paper demonstrated that (1) the

**Table 2. Mean $R^2$ in published TWAS projects.**

| Title of the publication | Description of prediction accuracy |
|---|---|
| Large-scale transcriptome-wide association study identifies new prostate cancer risk regions [22] | The mean $R^2$ = 0.07 for measured and predicted gene expression for TCGA normal prostate samples using models fitted in GTEx normal prostate. |
| A framework for transcriptome-wide association studies in breast cancer in diverse study populations [45] | The median CV $R^2$ for the 153 genes is 0.011 in both African American and white women. |
| Evaluation of PrediXcan for prioritizing GWAS associations and predicting gene Expression [46] | The average of prediction accuracy ($R^2$) is 0.023 for the DGN model and 0.02 for the GTEx model, with both using whole blood model. |
| A gene-based association method for mapping traits using reference transcriptome data [18] | The average prediction $R^2$ value is 0.0197 for GEUVADIS LCLs. For GTEx tissues, the prediction $R^2$ values are 0.0367 (adipose), 0.0358 (tibial artery), 0.0356 (left-ventricular heart), 0.0359 (lung), 0.0269 (muscle), 0.0422 (tibial nerve), 0.0374 (sun-exposed skin), 0.0398 (thyroid) and 0.0458 (whole blood). |

predictive accuracy of expressions is significantly increased, and (2) EpiXcan enabled the discovery of novel genes [27]. We present here the first rigorous power analysis of EpiXcan. We first conduct simulations where a subset of SNPs are assigned increased effects, which reflects the main insight of the EpiXcan paper that epigenetic-relevant functional SNPs have higher impact on variation in gene expression. In particular, we assume the real effect size follows a standard normal distribution $N(0,1)$, and sample effect sizes from this distribution. Assuming these functional SNPs are known (based on various techniques of annotating SNP functions), we relieve their penalty in training the predictive model. Using the predicted expressions, we calculate power using our derived NCP, and compare the resulting analysis with the standard TWAS protocol. **S1**–**S4 Figs** depict this quantitative evaluation of the improvement in power due to the contribution of epigenetic-relevant functional SNPs. Evidently, under the causality model EpiXcan indeed increases power by improving expression predictions, although the improvement is more pronounced in the cases that expression heritability is low (**S1 and S2 Figs**). However, under the pleiotropy model, EpiXcan only shows almost no increase in power over TWAS (**S3 and S4 Figs**). This observation suggests that when DNA mutations contribute to phenotype directly, the benefit of more accurate predictions for expressions may not be substantial. The source data for **Figs 2**–**5** and **S1**–**S4 Figs** are included in S1 Data.

## Discussion

In this work, we produced a novel derivation of the NCP for LMMs based on the decorrelation procedure, allowing us to calculate closed-form estimates of statistical power for three protocols: GWAS, emGWAS, and TWAS. Our power analysis revealed two practical insights. **First**, in the pleiotropy scenario, the use of predicted expressions in TWAS is overwhelmingly more powerful than the use of actual expressions in emGWAS, regardless of the accuracy of the predicted expressions *per se* (**Table 1**). This suggests that even if real expressions can be experimentally determined, TWAS is still superior for the analysis of some genes. While this appears counterintuitive, in statistical terms it is a direct result of the lack of a causal relationship between expression and phenotype under pleiotropy. This result reinforces the key insight, as presented by some publications [18], that TWAS uses expression as an objective function to select a linear combination of genetic variants, rather than attempting to accurately predict expressions. We note that this is equivalent to denoising in the field of machine learning [47]. **Second**, expression heritability determines the relative power of TWAS and GWAS. When the expression heritability is lower than 0.04 (in the casualty

scenario) or 0.06 (in the pleiotropy scenario), GWAS outperforms TWAS despite not utilizing gene expression information. This suggests that in practice, TWAS may often be suboptimal when expression heritability is low (**Table 2** and **Table** 1 in [18]), which can be mitigated by choosing the optimal association mapping protocol according to this work's quantitative guidelines.

A recent publication has also compared the statistical powers of GWAS and TWAS using pure simulations [33]. However, since we calculate power from a closed-form NCP derivation, our work establishes theoretical benchmarks for the performance of each protocol, independent of their implementations. Our work also has a different focus: rather than comparing techniques for training the genotype-expression predictive model and the impact of the actual number of causal genetic variants, we rank the effectiveness of GWAS, TWAS and emGWAS to better guide the practical application of TWAS. We analyze the theoretical effectiveness of real expressions as utilized by emGWAS, but exclude the protocol eGWAS as analyzed in [33], which uses eQTLs to assist association mapping. Our conclusions also differ slightly, as while the previous publication highlighted the importance of expression heritability, they concluded that expression heritability affects power only under the causality scenario, and not pleiotropy. In contrast, we concluded that expression heritability affects both scenarios.

Finally, our closed-form derivation is readily adaptable to other methods utilizing middle 'omics' (endophenotypes) such as IWAS [28] and PWAS [29,34]. In fact, the variable $Z$ in formula (15) can already represent such data as images or proteins, and thus no further modifications of the NCPs are necessary to adapt this work.

The present NCP framework only focuses on statistical power for detecting associations, and is not able to determine causality in the framework of Mendelian randomization such as in SMR and its extensions [48,49]. As a future work, we may attempt to derive closed-form power analyses for the MR framework.

There are several limitations in the present study. Although our closed-form derivation is easily adaptable and works independently of specific implementations, it is unable to capture power loss due to implementation limitations or bias in specific datasets. Additionally, closed-form derivations are more sensitive to model assumptions than simulation-based methods. Our calculation of the NCP also requires the variance component $\sigma_g^2$ to be estimated from data, in order to form the decorrelation matrix $D_x$. Although this approximation introduces extra variability and may therefore cause a decrease in power, we have omitted this variability from our analyses as the estimation of $\sigma_g^2$ is generally well-established, and has high accuracy in practice when given thousands of samples. Finally, we only compared linear models for GWAS and TWAS. As a future work, we may explore kernel-based nonparametric and semi-parametric methods for conducting both GWAS [50,51] and TWAS [52].

## Supporting information

**S1 Fig. Causality scenario of EpiXcan and TWAS when expression heritability is high and PVX is low.** The PVX (phenotypic variance explained by expression) is 0.005, 0.01, 0.025, and 0.05 in the four columns as indicated by the X-axis labels. In each of (a), (b), and (c), the expression heritability for the top and bottom rows are 0.1 and 0.3 respectively. The number of genes contributing to phenotype for (a), (b) and (c) are 4, 9, and 13 respectively. The number of causal variants per gene is randomly sampled from the interval [4,9].
(TIFF)

**S2 Fig. Causality scenario of EpiXcan and TWAS when expression heritability is low and PVX is high.** In each panel, the PVX is 0.05 and 0.1 in the left and right columns as indicated

by the X-axis labels. In each of (a), (b), (c), and (d), the numbers of genes contributing to phenotype for the left, center, and right panels are 4, 9, and 13 respectively. The expression heritability levels in (a), (b), (c), and (d) are 0.025, 0.03, 0.04, and 0.08 respectively. The number of causal variants per gene is randomly sampled from the interval [4,9].
(TIFF)

**S3 Fig. Pleiotropy scenario of EpiXcan and TWAS when expression heritability is high and trait heritability is low.** The trait heritability is 0.005, 0.01, 0.025, and 0.05 in the four columns as indicated by the X-axis labels. In each of (a), (b), and (c), the expression heritability for the top and bottom panels are 0.1 and 0.3 respectively. The numbers of genes contributing to phenotype for (a), (b), and (c) are 4, 9, and 13 respectively. The number of causal variants per gene is randomly sampled from the interval [4,9].
(TIFF)

**S4 Fig. Pleiotropy scenario of EpiXcan and TWAS when expression heritability is low and trait heritability is high.** The PVX is 0.05 and 0.1 in the two columns as indicated by the X-axis labels. In each of (a), (b), (c), and (d), the number of genes contributing to phenotype for the left, center, and right panels are 4, 9, and 13 respectively. The expression heritability levels in (a), (b), (c), and (d) are 0.025, 0.04, 0.06, and 0.08 respectively. The number of causal variants per gene is randomly sampled from the interval [4,9].
(TIFF)

**S1 Data. Source data for Figs 2–5 and S1–S4 Figs.** The six columns are scenarios, protocols, number of genes, expression heritability, trait heritability and power respectively.
(XLSX)

**S1 Text. Supplementary information and detailed mathematical derivations.**
(DOCX)

## Author Contributions

**Conceptualization:** Chen Cao, Bowei Ding, Jingjing Wu, Quan Long.

**Data curation:** Chen Cao, Bowei Ding, Qing Li, Devin Kwok.

**Formal analysis:** Chen Cao, Bowei Ding.

**Funding acquisition:** Jingjing Wu, Quan Long.

**Investigation:** Jingjing Wu, Quan Long.

**Methodology:** Chen Cao, Bowei Ding, Jingjing Wu, Quan Long.

**Project administration:** Jingjing Wu, Quan Long.

**Resources:** Chen Cao, Bowei Ding, Qing Li, Devin Kwok, Jingjing Wu, Quan Long.

**Software:** Chen Cao, Bowei Ding.

**Supervision:** Jingjing Wu, Quan Long.

**Validation:** Chen Cao, Bowei Ding.

**Visualization:** Chen Cao, Devin Kwok.

**Writing – original draft:** Chen Cao, Bowei Ding, Devin Kwok, Quan Long.

**Writing – review & editing:** Devin Kwok, Jingjing Wu, Quan Long.

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
