## [Decision Letter · Decision Letter 0]

5 Dec 2020

Dear Dr Long,

Thank you very much for submitting your Research Article entitled 'Power analysis of transcriptome-wide association study: implications for practical protocol choice' to PLOS Genetics.

The manuscript was fully evaluated at the editorial level and by independent peer reviewers. The reviewers appreciated the attention to an important topic but identified some concerns that we ask you address in a revised manuscript

We therefore ask you to modify the manuscript according to the review recommendations. Your revisions should address the specific points made by each reviewer.

We would like to see some detailed comparisons with other methods (Epixcan and SMR, reviewer 1) and address the independence of the outcome model and expression model (reviewer 2).

[LINK]

Yours sincerely,

Xiaofeng Zhu

Associate Editor

PLOS Genetics

David Balding

Section Editor: Methods

PLOS Genetics

Reviewer's Responses to Questions

**Comments to the Authors:**

Reviewer #1: In this article, authors provided power analysis of TWAS to indicate practical protocol choice. It is a very interesting topic and the results are pretty interesting. PrediXcan and EpiXcan are cited and the methods performed TWAS to identify trait-associated transcriptomes. Just one minor suggestion:

can authors compare Epixcan and the other related methods? Maybe another interesting paper using SMR could provide more insights: DOI: 10.1016/j.ajhg.2017.04.016. PMID: 28552197

Reviewer #2: In this manuscript, Ding and colleagues studies power of transcriptome-wide association studies (TWAS), providing practical guidelines based on results accordingly. Specifically, the authors compared power of TWAS with two alternative strategies: genome-wide association studies (GWAS) and association between measured gene expression levels with phenotype of interest (termed emGWAS by the authors). The comparisons are meaningful and the consideration of the emGWAS is thoughtful as such data may become increasingly available in the near future. The authors considered two main scenarios: causal or mediation scenario where genetic variants exert effect on phenotype via gene expression, and pleiotropic scenario where genetic variants simultaneously influence both phenotype and gene expression. There are many merits of the study, including analytical derivations to lay the theoretical foundation of power calculations, and carefully designed simulation studies to evaluate the power of TWAS and related approaches, which have clear implications in practice for real studies with or without measured expression data.

Generally speaking, the manuscript is well written with clearly presented methods and results.

I have the following specific questions and comments (all relatively minor) that can help further improve the manuscript.

(1) (Probably the only comment that is between major and minor): the authors make an independence assumption between the outcome model (4) and expression model (5), as explicitly stated in lines 243-244, and again in line 256. It is not clear to me what independence of two models mean exactly. It is also not clear what the consequences would be if this assumption is violated. The authors should at least make the former clear and discuss the latter.

(2) Lines 158-160, the authors stated “We calculate the GRM from DNA instead of expressions using the assumption that the ultimate goal is to identify genetic variants underlying expressions.” The logic here is awkward: the decision of using GRM based on genotypes is well warranted but not because the goal is to identify genetic variants underlying expression. To me, using genotypes to derive GRMs is justified because they provide reasonable and probably more accurate estimates of pairwise relationships among study participants than correlations based on predicted expression data.

**Have all data underlying the figures and results presented in the manuscript been provided?**

Reviewer #1: Yes

Reviewer #2: Yes

PLOS authors have the option to publish the peer review history of their article (what does this mean?). If published, this will include your full peer review and any attached files.

Reviewer #1: No

Reviewer #2: **Yes: **Yun Li

---

## [Decision Letter · Decision Letter 1]

6 Feb 2021

Dear Dr Long,

We are pleased to inform you that your manuscript entitled "Power analysis of transcriptome-wide association study: implications for practical protocol choice" has been editorially accepted for publication in PLOS Genetics. Congratulations!

Yours sincerely,

Xiaofeng Zhu

Associate Editor

PLOS Genetics

David Balding

Section Editor: Methods

PLOS Genetics

Comments from the reviewers (if applicable):

Reviewer's Responses to Questions

**Comments to the Authors:**

Reviewer #1: In the revisions, authors provided a good amount of analyses to verify the power of the method that proposed. For instance, analysis to compare the performance of other methods, say, EpiXcan. The results are convincing, which further make the paper complete. I think the paper is acceptable now.

Reviewer #2: The authors have carefully addressed all my comments. I have no further comments. The authors should be congratulated on presenting this very useful piece of work!

**Have all data underlying the figures and results presented in the manuscript been provided?**

Reviewer #1: Yes

Reviewer #2: None

PLOS authors have the option to publish the peer review history of their article (what does this mean?). If published, this will include your full peer review and any attached files.

Reviewer #1: No

Reviewer #2: **Yes: **Yun Li

**Data Deposition**

http://datadryad.org/submit?journalID=pgenetics&manu=PGENETICS-D-20-01247R1

**Press Queries**

---

## [Editor Report · Acceptance letter]

19 Feb 2021

PGENETICS-D-20-01247R1 

Power analysis of transcriptome-wide association study: implications for practical protocol choice 

Dear Dr Long, 

We are pleased to inform you that your manuscript entitled "Power analysis of transcriptome-wide association study: implications for practical protocol choice" has been formally accepted for publication in PLOS Genetics! Your manuscript is now with our production department and you will be notified of the publication date in due course.

With kind regards,

Alice Ellingham

PLOS Genetics

On behalf of:
